# The relative abundance of languages: Neutral and non-neutral dynamics

Luís Borda-de-Água [1,2,3]*, Stephen P. Hubbell[4,5]

1 CIBIO/InBio, Centro de Investigação em Biodiversidade e Recursos Genéticos, Laboratório Associado, Universidade do Porto, Vairão, Portugal, 2 CIBIO/InBio, Centro de Investigação em Biodiversidade e Recursos Genéticos, Laboratório Associado, Instituto Superior de Agronomia, Universidade de Lisboa, Lisboa, Portugal, 3 BIOPOLIS Program in Genomics, Biodiversity and Land Planning, CIBIO, Vairão, Portugal, 4 Department of Ecology and Evolutionary Biology, University of California, Los Angeles, California, United States of America, 5 The Smithsonian Tropical Research Institute, Balboa, Panama

* lbagua@cibio.up.pt

**Data Availability Statement:** The data supporting the results of your study can be found here: Gordon RG, Grimes BF (ed.). Ethnologue: Languages of the World. Dallas, Tex.: SIL International; 2005. Interested researchers can replicate this study using the protocols listed in the

## Abstract

Credible estimates suggest that a large number of the nearly 7000 languages in the world could go extinct this century, a prospect with profound cultural, socioeconomic, and political ramifications. Despite its importance, we still have little predictive theory for language dynamics and richness. Critical to the language extinction problem, however, is to understand the dynamics of the number of speakers of languages, the dynamics of language abundance distributions (LADs). Many regional LADs are very similar to the bell-shaped distributions of relative species abundance predicted by neutral theory in ecology. Using the tenets of neutral theory, here we show that LADs can be understood as an equilibrium or disequilibrium between stochastic rates of origination and extinction of languages. However, neutral theory does not fit some regional LADs, which can be explained if the number of speakers has grown systematically faster in some languages than others, due to cultural factors and other non-neutral processes. Only the LADs of Australia and the United States, deviate from a bell-shaped pattern. These deviations are due to the documented higher, non-equilibrium extinction rates of low-abundance languages in these countries.

## Introduction

Linguistic richness, defined as the number of languages, is not evenly distributed on Earth [1, 2], with the majority of the language-rich countries situated in the tropics (e.g., [3]). Likewise, the number of speakers is not evenly distributed among languages; some languages have hundreds of millions of speakers while others only a few [4]. Similar patterns of richness and abundance are found in ecology. One of the recent advances in ecology has been the development of the Neutral Theory of Biodiversity and Biogeography [5] (hereafter NTBB) to explain patterns of relative species abundance, as measured for example by the number of individuals, on local to global scales. One prediction of NTBB theory is the patterns of relative species abundance to expect under the assumption of a dynamic equilibrium between the origin and extinction of species. The theory predicts that the steady-state distribution of species abundances will be bell-shaped on a logarithmic scale when species undergo fission into daughter

methods section. The authors had no special access to this dataset.

**Funding:** LBA was financed through Portuguese national funds through FCT – Fundação para a Ciência e a Tecnologia, I.P., under the Norma Transitória - DL57/2016/CP1440/CT0022. This work was supported by National Funds through FCT-Fundação para a Ciência e Tecnologia- in the scope of the project UIDB/50027/2020.

**Competing interests:** The authors have declared that no competing interests exist.

species. In the case of languages, when we consider their "abundance" measured by the number of speakers, the corresponding logarithmic-scale language abundance distributions (hereafter LADs) are also almost always bell-shaped, whether the sampling unit is a country or the entire world [6]; see also Fig 1. Furthermore, subjects in ecological and linguistic communities are subjected to similar processes, such as, birth, death, and probability of speciating (ecology) or giving rise to a new language (linguistics). Therefore, the similarities between the patterns predicted by the NTBB and language abundance distributions, and the similarity of the underlying processes, make the NTBB a natural candidate to test hypotheses about when language dynamics might be governed by neutral or non-neutral processes.

Previously, Dixon [8] has suggested a qualitative model of linguistic equilibrium, i.e., constant number of languages over time, where the unit was the political group. This is

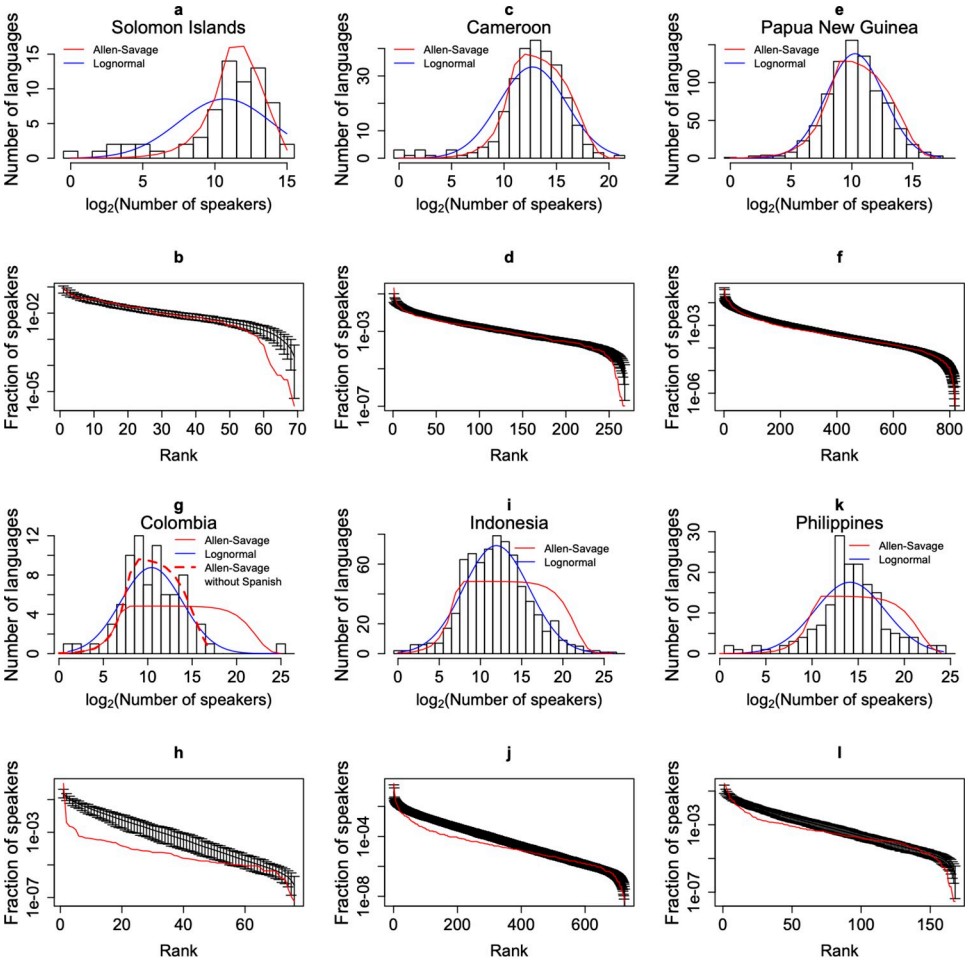

**Fig 1. Language abundance distributions.** Plots (a,c,e,g,i,k) show the histograms of the language abundance distributions and the fitted Allen-Savage (red line) and lognormal (blue line) distributions. The bins are centered in integer numbers, *n*, and have borders at *n*±0.5 (see, [7]). Plots (b,d,f,h,j,l) show the observed (red line), and fitted (black line) rank abundance distributions. Languages are ranked from the most abundant language on the left-hand side of the *x*-axis to the least abundant on the right-hand side. The errors bars correspond to 95% confidence intervals and were obtained for each country by sampling 200 times a number of points equal to the number of languages from a distribution with parameters corresponding to the maximum likelihood estimates. The Solomon Islands, Cameroon and Papua New Guinea, provide examples of very good fittings (a,b,c,d,e,f). Colombia is the typical example of a country with a dominant language, in this case Spanish, and yields a poor fit (g,h). The fitting improves considerably when we remove Spanish (red dashed line in plot g). Indonesia and Philippines are examples of poorly fitting curves that have a large plateau at intermediate language abundance classes (i,j,k,l).

reminiscent of the Theory of Island Biogeography [9] in which the unit was the species. In these cases, the neutrality is at the species, or political group, level, meaning they all have equal functional properties relevant to the dynamics of languages. In contrast, in NTBB, including our application to languages, the unit is the individual (or speaker) and the assumption is that all individuals are equal in their per capita rates of reproducing (transmitting the language to the offspring), dying and giving rise to a new language, the latter known as glossogenesis.

There are several versions of the NTBB depending on the speciation (or glossogenesis) mode adopted [10, 11]; and see (S3 Appendix). For present purposes, we assume that each incipient language (within a homogeneous region) starts with the same fixed number of individuals. Clearly, the assumption of the same starting number of individuals is a first approximation, but in broad agreement with previous estimates of initial population sizes [12]. Under this assumption, Allen and Savage [13] derived the equilibrium relative abundance distribution of the NTBB. An equilibrium relative abundance distribution means that the number of languages and the number of speakers distributed among languages is the same over time, implying that the total number of individuals is constant. However, this equilibrium is a dynamic steady-state, and does not imply that the languages in a community are always the same: some may go extinct and new ones may arise. Nor does it assume that languages always have the same number of speakers: some languages may increase in number of speakers while others may decrease in numbers.

The two parameters of the Allen-Savage distribution are $J_S$, the incipient size of the population, and $v$, the per capita glossogenesis (speciation) rate, both capturing important characteristics of language dynamics. In fact, estimating these parameters can help identify regions in the globe with different rates of species origination and the typical size of a human group. This observation is important because the bell-shape of the LADs, when plotted in a logarithmic scale, makes the lognormal a natural candidate to fit these distributions. However, the parameters of the lognormal distribution, the mean and variance, do not have any process interpretation whereas the parameters of the distribution suggested by Allen and Savage are readily interpreted mechanistically. However, for comparison with other published accounts, we will also fit the data using the lognormal distribution.

Although the assumption of equality at individual level has been controversial in ecology (e.g., [14]), the neutrality assumption of all language speakers is less likely to generate controversy, under the assumption that all individuals are broadly subject to the same social and environmental conditions. This in line with the applications of NTBB in ecology where it is assumed that all individuals belong to the same trophic level and the same biogeographic region (e.g., the Amazon basin). Despite being less controversial, the Allen-Savage distribution does not provide a good fit in all situations. This is likely due to non-neutral processes, in particular differential growth in the number of speakers among different languages. Indeed, although the human population has been growing globally in the last centuries, not all languages have been growing at the same rate, in fact, some have disappeared. What is remarkable is that non-neutral growth differences among some linguistic groups lead to clear departures from the LADs predicted by the Allen-Savage distribution, allowing us to distinguish which LADs are consistent with neutrality and which are not.

## Methods

### The model

Here we treat each country as a self-contained unit where the processes of death, birth and glossogenesis occur; thus, a country is the equivalent of a biogeographic region in the original

NTBB. The purpose of choosing country as the unit of analysis (and not as, say, continental or global scale) is to ensure that individuals are subject to similar conditions. Clearly, countries are not closed systems; migrations occur and often the same language is spoken in different countries. The advantage of using a country as the basic spatial unit is that we consider, as a first approximation, that its linguistic populations are subjected to the similar environmental and social conditions. On the other hand, we assume that if migrations occur these do not have a strong impact in the overall language abundance distribution; if this is not the case, such is the case in periods of social upheaval, the model we now describe will not apply.

Allen and Savage [13] derived the abundance distribution in the case of a biogeographic region under the assumptions of the NTBB and by supposing that an incipient language starts with a constant number of individuals. When applied to language dynamics the assumptions of the Allen-Savage model for a given country are that (i) the average number of speakers of an incipient language is $J_S$, (ii) the total population of speakers of all languages, $J$, in the country of interest is very large relative to $J_S$, (iii) the total origination rate of languages equals the extinction rate, (iv) different populations have similar per capita rate of glossogenesis and (v) the total population, $J$, fluctuates stochastically around a mean value. Some of these assumptions are unrealistic to present human populations. For instance, assumption (iii) implies that the number of languages is constant over time, and we know that at present a large number of languages is becoming extinct. Equally, assumption (v) does not hold at the present given the fast growth of most human populations. We will discuss their broad implications in due time.

## The Allen-Savage distribution

The two parameters of the Allen-Savage model to be estimated from the data are the size of the population speaking a new language, $J_S$, or the fraction $P_S = J_S/J$, and the rate of origination of languages, glossogenesis, $v$, which is usually combined with the total population size of the country, $J$, to form the parameter $\theta = 2Jv$. The parameter $\theta$ is a dimensionless language diversity number, corresponding to the fundamental biodiversity number of NTBB, that reflects the fact that the richness of languages is a function of the per capita rate of glossogenesis, $v$, and of the total size of the population, $J$. We use likelihood methods to estimate the parameters $P_S$ (or $J_S$) and $\theta$ (or $v$). However, the exact likelihood formula [15] (Etienne and Alonso, 2005) is computationally demanding, therefore, we use an approximate likelihood formula [10, 16] based on the following considerations. Consider the probability of a language having $N$ speakers, $p(N)$ in a population of $J$ speakers of all languages. Probability $p(N)$ can be estimated from the ratio of the expectation of the number of languages with $N$ speakers, $E(L_N|J)$ to the expectation of the language richness, $E(L|J)$,

$$p(N) = \frac{E(L_N|J)}{E(L|J)}.$$

Then the likelihood function, $\ell$, is [10, 16]

$$\ell = \frac{L!}{\prod\limits_{N=1}^{J} L_N} \prod\limits_{N=1}^{J} p(N)^{L_N}.$$

Under the assumption that an incipient language starts with a constant number of individuals, $J_S$, for a country with $J$ speakers distributed among $L$ languages, the probability, $p(N)$, of

finding languages with $N$ speakers is [13]

$$p(N) = \begin{cases} \dfrac{1 - \exp(-\theta N/J)}{[\gamma + \exp(\theta P_S)E_1(\theta P_S) + \log(\theta P_S)]N} & N < J_S \\[2em] \dfrac{[\exp(\theta P_S) - 1]\exp(-\theta N/J)}{[\gamma + \exp(\theta P_S)E_1(\theta P_S) + \log(\theta P_S)]N} & N \geq J_S \end{cases} \tag{1}$$

where $\gamma \approx 0.57721$ is the Euler-Mascheroni's constant, and $E_1(x)$ is the exponential integral function [17]. We refer to this distribution as the "Allen-Savage distribution".

## The data

We used data on languages and number of speakers per country from the Ethnologue [4]. The reason for choosing country as the unit of analysis (and not, say, continental or global scales) is to ensure that individuals are subject to similar conditions. However, this leaves the question of the environmental heterogeneity within a country, and how this heterogeneity affects the in-country linguistic diversity, unanswered. Obviously, contingent historical events determine the richness of languages but environmental factors are also likely to play a key role. According to Nettle [1], language richness is a function of the ecological risk, especially is non-industrial societies. By ecological risk it is meant the degree by which human populations are exposed to the vagaries of their natural environment. The justification is that when ecological risk is high, human populations are more dependent on each other and, thus, local language differences do not diverge to the point of becoming mutually unintelligible. On the other hand, if ecological risk is low populations are less dependent on each other and local language variations can more easily diverge forming new languages. Therefore, the higher the ecological risk the smaller the number of languages. In order to measure ecological risk, Nettle [1] used the mean growing season, defined as the number of months in which the monthly rainfall (in millimeters) is greater than twice the monthly temperature ([1], p. 82). Although we acknowledge that this is a very simple measure of ecological risk, of the environmental determinants of linguistic diversity (but see, [18–20], we will use it as a first approximation to guarantee relatively homogeneous regional units. This reduces the number of countries to 46 (i.e., those with standard deviation of the growing season smaller than two months) and to a total of 4099 languages.

## Results

Using the above likelihood method, we obtained values for parameters $P_S$ (or $J_S$) and $\theta$ (or $\nu$). Table 1 shows the results for countries with more than 50 languages; (S1 Appendix) shows the results for all countries studied. In Fig 1 we show examples of fitted distributions; and in (S1 Appendix) we show the fitting to all countries. We distinguish two situations: those for which the Allen-Savage distribution is bell-shaped and gives a good fit, Fig 1A–1F (Solomon Islands, Cameroon and Papua New Guinea), and those for which the Allen-Savage distribution exhibits a plateau at intermediate language abundances and gives a poor fit, Fig 1G–1L (Colombia, Indonesia and Philippines). As previously mentioned, we also fitted the language abundance distributions with the lognormal distribution (blue line in Fig 1). Unlike the Allen Savage distribution, the lognormal distribution never exhibits a plateau. To assess the fit of the Allen-Savage and lognormal distributions, we used the ratio of the Akaike weights [21]. Excluding Colombia and the Asian countries for which the Allen-Savage distribution gives a poor fit, in most cases there is not a clear best distribution (Table 2). When there is a best distribution, it is usually the Allen-Savage distribution (e.g., Cameroon). A notable exception is Papua New

**Table 1. List of countries with more than 50 languages, their number of languages, number of individuals, $J$, and the maximum likelihoods of $\theta = 2Jv$, $v$ (the glosso-genesis rate), $P_s$ (the fraction of the number of individuals, relatively to $J$, of an incipient language), and $J_s = P_s * J$.** Countries with name in italic correspond to cases of poor fitting, as revealed by the rank abundance plots. See (S1 Appendix) for the complete list of countries.

| Continent | Country | Number of Languages | $J$ (individuals) | $\theta$ | $v$ | $P_S$ | $J_S$ |
|---|---|---|---|---|---|---|---|
| Africa | *Benin* | 53 | 6449442 | 17 | 1.32E-06 | 4.60E-03 | 2.97E+04 |
| Africa | Congo | 57 | 3366116 | 12 | 1.78E-06 | 1.10E-03 | 3.70E+03 |
| Africa | Burkina Faso | 64 | 11018638 | 10 | 4.54E-07 | 2.20E-04 | 2.42E+03 |
| Africa | Cent. Afr. Republic | 67 | 3298745 | 16 | 2.42E-06 | 1.40E-03 | 4.62E+03 |
| Africa | Ghana | 73 | 20124060 | 11 | 2.73E-07 | 2.20E-04 | 4.43 E+03 |
| Africa | Côte d'Ivoire | 77 | 9150469 | 15 | 8.20E-07 | 5.50E-04 | 5.03E+03 |
| Africa | Tanzania | 113 | 25687658 | 28 | 5.45E-07 | 1.10E-03 | 2.83E+04 |
| Africa | Chad | 123 | 5809406 | 21 | 1.81E-06 | 2.30E-04 | 1.34E+03 |
| Africa | Dem. Rep. Congo | 202 | 38399510 | 35 | 4.56E-07 | 1.40E-04 | 5.38E+03 |
| Africa | Cameroon | 268 | 9637152 | 56 | 2.90E-06 | 2.30E-04 | 2.22E+03 |
| America | *Colombia* | 76 | 34571380 | 7 | 1.01E-07 | 4.10E-06 | 1.42E+02 |
| Asia | *Thailand* | 62 | 53529352 | 6 | 5.60E-08 | 5.60E-06 | 3.00E+02 |
| Asia | Laos | 80 | 5300189 | 12 | 1.13E-06 | 1.90E-04 | 1.01E+03 |
| Asia | *Vietnam* | 92 | 75273638 | 8 | 5.31E-08 | 3.30E-06 | 2.48E+02 |
| Asia | *Myanmar* | 98 | 46606310 | 14 | 1.50E-07 | 1.00E-04 | 4.66E+03 |
| Asia | *Nepal* | 117 | 22813243 | 16 | 3.51E-07 | 5.20E-05 | 1.19E+03 |
| Asia | *Malaysia* | 125 | 15566135 | 15 | 4.82 E-07 | 2.30E-05 | 3.58E+02 |
| Asia | *Philippines* | 168 | 70626954 | 20 | 1.42E-07 | 2.00E-05 | 1.41E+03 |
| Asia | *India* | 397 | 945679579 | 38 | 2.01E-08 | 1.30E-06 | 1.23E+03 |
| Asia | *Indonesia* | 724 | 218610076 | 70 | 1.60E-07 | 6.90E-07 | 1.51E+02 |
| Oceania | Solomon Islands | 69 | 353992 | 29 | 4.10E-05 | 6.70E-03 | 2.37E+03 |
| Oceania | Vanuatu | 109 | 117494 | 30 | 1.28E-04 | 1.45E-03 | 1.70E+02 |
| Oceania | Papua New Guinea | 819 | 3665383 | 193 | 2.63E-05 | 1.20E-04 | 4.40E+02 |

Guinea, where the lognormal provides a better description; nevertheless, visual inspection and the confidence interval plot, Fig 1F, do not reveal a clear advantage to the lognormal.

We urge caution when interpreting the values of $J_S$ and $v$ (Table 1) because these values are estimated from countries where populations have been growing. Relating these estimated values of $J_S$ with those of the typical size of an ethnic group (e.g., [22]) originating a language would give a wrong estimate. As we discuss in the next section, if all populations grow at the same rate, what remains constant is $P_S = J_S/J$ and $\theta = 2vJ$, and any attempts to relate the values of $J_S$ and $v$ with any real attributes of the populations have to consider the total size of the populations at a point in time when those populations were under the equilibrium assumptions of the Allen-Savage distribution; see also (S4 Appendix).

## Discussion

We introduced a model to describe the relative abundance of languages, as measured by their number of speakers, based on the neutral theory of biodiversity and biogeography [5]. We assumed that languages arise with a probability of origination $v$ and specific number of speakers $J_S$; under these assumptions the relative abundance of languages is described by the Allen-Savage distribution. (We also explored the fitting of the languages distributions under the assumption of variable $J_S$ but with poor results; see (S2 Appendix)). One of the strengths of our approach is that the parameters of the Allen-Savage distribution are readily interpreted in terms of the probability of a new language arising, $v$, and the size of the its population at the origin, $J_S$. The estimation of these parameters using realistic (total) population sizes may help

**Table 2. Corrected Akaike Information Criterion values (AICc) for the Allen-Savage (AS) and lognormal (logn) distributions, their weights (*w*) (Burnham and Anderson 2010) and their ratio, $w_{AS}/w_{logn}$, for the countries with more than 50 languages.** See (S1 Appendix) for the complete list of countries.

| Country | AICc AS | AICc logn | $w_{AS}$ | $w_{logn}$ | $w_{AS}/w_{logn}$ |
|---|---|---|---|---|---|
| Benin | 1332.3 | 1333 | 5.915E-01 | 4.085E-01 | 1.448E+00 |
| Congo | 1310.6 | 1307.8 | 2.010E-01 | 7.990E-01 | 2.516E-01 |
| Burkina Faso | 1581.1 | 1572.2 | 1.166E-02 | 9.883E-01 | 1.180E-02 |
| Cent. Afr. Republic | 1567.3 | 1576.2 | 9.885E-01 | 1.154E-02 | 8.563E+01 |
| Ghana | 1855.8 | 1853.7 | 2.592E-01 | 7.408E-01 | 3.499E-01 |
| Côte d'Ivoire | 1899.2 | 1898.4 | 4.110E-01 | 5.890E-01 | 6.977E-01 |
| Tanzania | 2975.4 | 2983.5 | 9.832E-01 | 1.679E-02 | 5.856E+01 |
| Chad | 2783.1 | 2790.8 | 9.790E-01 | 2.104E-02 | 4.653E+01 |
| Dem. Rep. Congo | 5051.9 | 5050.7 | 3.475E-01 | 6.525E-01 | 5.326E-01 |
| Cameroon | 5905 | 5918.2 | 9.987E-01 | 1.305E-03 | 7.651E+02 |
| Colombia | 1505.3 | 1450.3 | 1.117E-12 | 1.000E+00 | 1.117E-12 |
| Thailand | 1532.1 | 1522 | 6.181E-03 | 9.938E-01 | 6.220E-03 |
| Laos | 1769 | 1759.1 | 7.034E-03 | 9.930E-01 | 7.083E-03 |
| Vietnam | 2170.4 | 2142.5 | 8.777E-07 | 1.000E+00 | 8.777E-07 |
| Myanmar | 2460.4 | 2436.6 | 6.676E-06 | 1.000E+00 | 6.676E-06 |
| Nepal | 2693.1 | 2674.7 | 9.903E-05 | 9.999E-01 | 9.904E-05 |
| Malaysia | 2690.8 | 2664.8 | 2.283E-06 | 1.000E+00 | 2.283E-06 |
| Philippines | 4134.2 | 4094.6 | 2.492E-09 | 1.000E+00 | 2.492E-09 |
| India | 10474.8 | 10408.8 | 4.613E-15 | 1.000E+00 | 4.613E-15 |
| Indonesia | 15709.1 | 15504.8 | 4.163E-45 | 1.000E+00 | 4.163E-45 |
| Solomon Islands | 1320.9 | 1333.4 | 9.980E-01 | 2.005E-03 | 4.977E+02 |
| Vanuatu | 1709.1 | 1722.4 | 9.988E-01 | 1.242E-03 | 8.043E+02 |
| Papua New Guinea | 14800.1 | 14784.7 | 4.526E-04 | 9.995E-01 | 4.528E-04 |

identify regions where large number of languages arose and the of the typical size of ethnolinguistic group originating a new language, and how these parameters vary among different regions of the globe. However, the interpretation of the numerical values of *v* and $J_S$ warrants some considerations because human populations are not in the equilibrium conditions assumed by the Neutral model, as we now discuss; see also (S4 Appendix).

When analyzing the dynamics of human populations an unavoidable consideration is their (very fast) growth in recent centuries, in particular, in the 20[th] century. This is in clear contradiction to one of the assumptions of the Allen-Savage distribution. Therefore, exploring the implications of growth to explain the failure of the Allen-Savage distribution to fit some distributions is in order. Moreover, we should not expect all populations to grow at the same rate, especially, if populations are identified by an attribute, such as language. In fact, among some linguistic groups the number of speakers has decreased, eventually to the point of extinction. This does not necessarily imply that, when the number of speakers decreases, the individuals representing those speakers are no longer present in the community. They may have been eliminated, as in the case of genocide, but in other cases speakers may have been forced, or they may have made a voluntary shift, from one language to another, as when, for instance, parents do not teach their children their mother tongues and instead adopt another language perceived as having more prestige or bringing more socio-economic benefits, such as access to education.

The development of a plateau for some Allen-Savage curves is an important result of this work, and we argue that this is a signature of non-neutral dynamics. Among LADs with a plateau one can distinguish two distinct potential mechanisms of non-neutral dynamics. One

mechanism is exemplified by Colombia (Fig 1G), the other by Indonesia and Philippines (Fig 1I and 1K). In Colombia´s case there is one language (Spanish) that has a much larger number of speakers than all the rest; see also the LADs of Myanmar and Vietnam in the (S1 Appendix). If we remove the largest language, then the Allen-Savage distribution gives a good fit to the remaining distribution. This example also serves to show that the maximum likelihood estimators of the Allen-Savage distribution are affected by extreme values, while those of the lognormal distribution are not. In the situation illustrated by Indonesia and Philippines (Fig 1I–1K) there is not a language with a much larger (isolated) number of speakers, like Spanish in Colombia's LAD, but nevertheless the Allen-Savage distribution still exhibits an extended plateau. The obvious departure of a bell-shaped liked distribution is particularly obvious for Philippine's LAD, where, in fact, even the lognormal gives a poor fit.

We interpret the results of the previous paragraph as revealing the effects of recent differential populations growth rates. In the following we present an explanation for the plateau exhibited by some fitting distributions. First, consider the situation in which, in a given geographical region the, number of speakers of each language grows at the same rate as all other languages. For simplicity, assume that the period of observation of the growth of languages is short, so that no new languages emerge. Also assume that at the beginning of the observation period, $t = 0$, the size of the linguistic groups is at the equilibrium given by Eq 1. Now let these populations collectively start to grow. If all populations have the same growth rate, then the number of speakers of a language $i$, at a time $t>0$, $N_{it}$, relates to the number at time $t = 0$, $N_{i0}$, as $N_{it} = C_t N_{i0}$, where $C_t$ is the same for all populations. In this case the total population size at time $t$ is $J_t = C_t J_0$. In terms of the histograms of the log transformed values of the number of speakers, such as depicted in Fig 1, the growth of the populations at the same rate corresponds to a shift by $\log_2(C_t)$ of the distributions to the right in the $x$-axis because $\log_2(N_{it}) = \log_2(C_t) + \log_2(N_{i0})$. Since the shift, $\log_2(C_t)$, is the same for all populations, only the mean of the distribution changes but not its variance. Notice that the parameters $\theta$ and $P_s$ of Eq 1 are the same, because a change of variable from $N$ to $P = N/J$ does not change the analytic expression of Eq 1, and the distributions for $t \neq 0$ will have different rate of glossogenesis, $v$, and incipient population size, $J_S$; See (S2 Appendix). In other words, the parameters $v$ and $J_S$ estimated from a group of populations that have been growing are different from those of the same populations when their sizes are in equilibrium, but $\theta$ and $P_s$ are the same.

Now consider a different situation, one in which languages have different growth rates. Using the same notation as before, $N_{it} = C_{it} N_{i0}$, but $C_{it}$ is no longer the same for all languages, hence the index $i$. The corresponding log transformed values are $\log_2(N_{it}) = \log_2(C_{it}) + \log_2(N_{i0})$. When $\log_2(C_{it})$ is not the same for all languages, the shift they experience along the $x$-axis when $t$ increases is not the same, and the resulting LAD does not have the same shape as the original distribution. What remains to be shown is the implication of differential growth to the fit provided by the Allen-Savage distribution. For simplicity, assume that $C_i>1$ (all populations grow) and assume that larger languages have an advantage over smaller ones, that is, larger languages have larger $C_i$. For the sake of example, assume that $C_{it} = \exp(r_i t)$ and that $r_i$ is proportional to the logarithm of the number of speakers at $t = 0$, $r_i = D\log(N_{i0})$, where $D$ is a positive constant (this does not have to be the case; it is just to obtain an simpler mathematical expression). Then

$$\log(N_{it}) = \log[N_{i0}\exp(D\log(N_{i0})t)] = (Dt + 1)\log(N_{i0}),$$

leading to distributions shifting to the right and, simultaneously, becoming wider as $t$ increases. The important point is that if we attempt to fit these distributions with the Allen-Savage distribution, we observe that, as time increases, the fitting curves develop a plateau. To

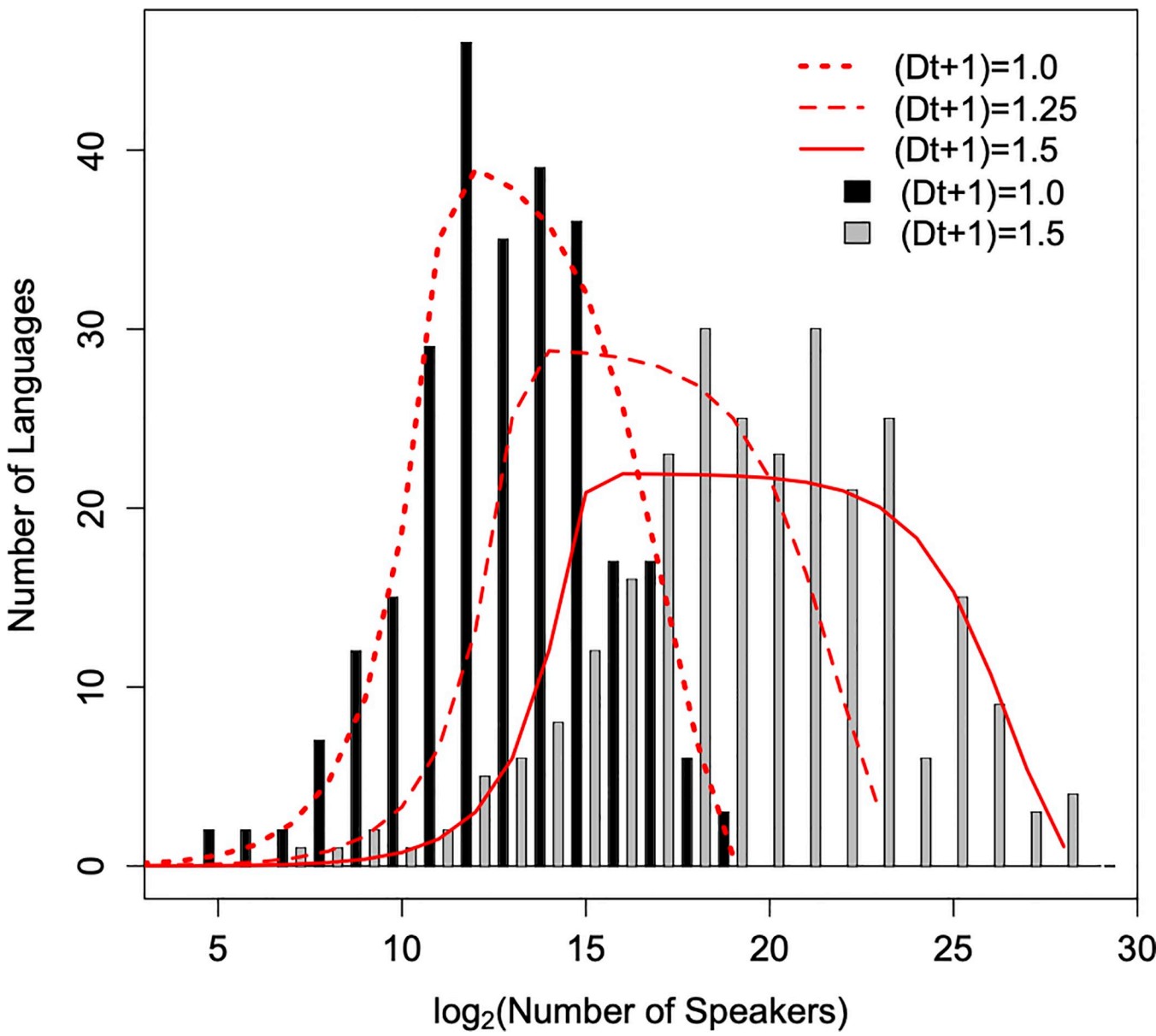

**Fig 2. Development of a plateau in the Allen-Savage distribution.** The black bars are the histograms of 268 points sampled randomly from an Allen-Savage distribution with parameters $\theta = 56$ and $P_S = 2.3\times10^{-4}$ (the same parameters as Cameroon). These data points were then multiplied by $(Dt+1) = 1.25$ and 1.5, the latter being shown in grey bars (bars for $(Dt+1) = 1.25$ not shown). The lines are the fits of Allen-Savage distributions obtained with likelihood methods. Notice the development of a plateau for intermediary abundance classes when $(Ct+1)$ increases. The bins are centered in integers numbers, $n$, and have borders at $n$ ±0.5.

illustrate this, we use the LAD of Cameroon, a distribution that is well fitted by the Allen-Savage distribution. It we allow its languages to growth at different rates, the fitted Allen-Savages distributions develop a plateau that becomes more pronounced as $t$ increases, as illustrated in Fig 2. Note that the LAD of Cameroon at $t = 0$ is the real one; thus the plateau of the fitted distribution depicted in Fig 2 could be a prediction of our model if the Cameroon languages were to start growing at different rates.

In summary, if a LAD is initially described by the Allen-Savage distribution and the populations start growing at approximately the same rate (neutral growth), then the resulting LADs

are well fitted by the Allen-Savage distribution, with the same $\theta$ and $P_S$ values. On the other hand, if populations have differential (non-neutral) growth rates, then the fitted Allen-Savage distributions develop a plateau at intermediate language abundances that widens over time as populations grow. Therefore, a plausible explanation for the plateau in Fig 1G, 1I and 1K, see (S1 Appendix) is the differential, non-neutral, growth of languages.

Two observations from our results are worth reporting. The first is that most LADs of African countries are well fitted by the Allen-Savage distribution; see (S4 Appendix). According to our previous discussion this could be explained by neutral (or near-neutral) growth among African linguistic groups. It is outside the scope of this work to identify the causes for such non-differential (neutral) growth, but these are likely to lie in the degree of centralization of political power or the enforcement of a few selected languages in education, usually those languages having higher prestige or spoken by larger ethnic groups. The second observation is that only Australia and the United States do not have bell shaped LADs, which are, instead, truncated bell-shaped distributions, being almost monotonically decreasing curves (Fig 3). However, these LADs are not evidence against the generality of the bell-shape pattern that arises under steady state conditions of language origination and extinction. Indeed, patterns such as those in the U.S.A. and Australia can arise from former LAD distributions that were once bell-shaped but have subsequently been modified by processes causing the number of speakers of the majority of languages to decline. Possible processes include forced or voluntary language shift to higher status languages, population declines due to European-introduced diseases, and genocide [23]. These processes shift the mode of the LAD distribution towards lower abundances, resulting in the observed truncated LAD distributions. What distinguishes the United States and Australia is that a large number of languages with a small number of speakers still remains, although many of these low-abundance languages are on the verge of extinction, thus the observed LADs are likely to represent a short transient period.

Finally, some considerations on the use of the lognormal distribution are in order. Previous work on language abundance distributions [24–30] emphasized the lognormal. Although the lognormal provides reasonably good fits, there is no demographic interpretation of the parameters, so that fitting a lognormal does not lead to further hypotheses to test. In contrast, all parameters of neutral theory applied to language abundance dynamics have demographic interpretations that generate testable hypotheses.

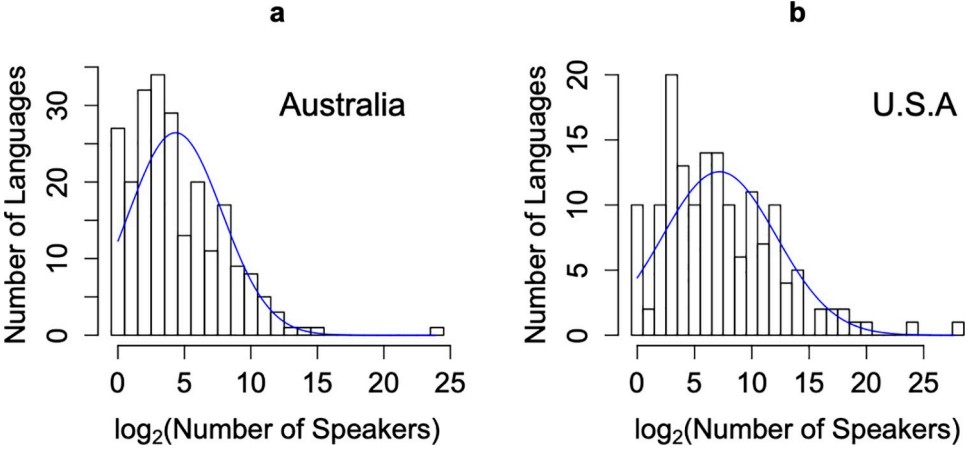

**Fig 3. Truncated language abundance distributions.** (a) Australia (232 languages) and (b) United States (156 languages). Among all the countries studied, these were the only distributions that did not conform to the bell-shaped pattern. These skewed distributions reflect the decreasing sizes and higher extinction rates of low-abundance languages in these countries. The blue curves are the best-fit lognormal distributions.

## Conclusions

We developed a new theory of the dynamics of languages that includes both origination and extinction under either equilibrium or non-equilibrium stochastic processes. We show that some language abundance distributions exhibit near-neutral dynamics, whereas others exhibit non-neutral dynamics. There are sufficient parallels between species and linguistic groups to suggest that a theoretical perspective similar to that developed by the NTBB in ecology might be useful in understanding the dynamics of language abundances that shape language diversity.

An important aspect of our approach is that we considered the relative number of speakers as a major determinant of linguistic diversity. We argue that any attempt to describe the dynamics of a system, or to identify causal relationships among its patterns and processes, that do not consider the relative abundance of its constituents is likely to miss an important determinant of its behavior; see also [5].

We anticipate that further development of a theory for language diversity will generate a wealth of testable hypotheses on language diversity and the underlying environmental and societal processes driving language dynamics, and it will bring changes in the respect for and protection of minorities' languages and cultures.

## Supporting information

**S1 Appendix. Additional information on the 46 countries analyzed.** Here we present results for all the countries analyses, see S1 Fig and S1-S3 Tables.
(DOCX)

**S2 Appendix. Population growth and the Allen-Savage distribution.**
(DOCX)

**S3 Appendix. The random fission and the protracted origination modes.**
(DOCX)

**S4 Appendix. Statistics for θ and the predicted origination rate ν' assuming a past period of total population size in equilibrium.**
(DOCX)

## Acknowledgments

We thank C. Capinha and H. M. Pereira for discussions on several matters concerning this paper, T. Hagemeijer, R. Mace and D. Nettle for discussions on linguistics and X. Chen and T. S. Ferguson for their advice on probability theory.

## Author Contributions

**Conceptualization:** Luís Borda-de-Água, Stephen P. Hubbell.

**Formal analysis:** Luís Borda-de-Água, Stephen P. Hubbell.

**Investigation:** Luís Borda-de-Água, Stephen P. Hubbell.

**Writing – original draft:** Luís Borda-de-Água.

**Writing – review & editing:** Luís Borda-de-Água, Stephen P. Hubbell.

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
