## [Decision Letter · Decision Letter 0]

27 Jul 2021

PONE-D-21-15698

The relative abundance of languages: neutral and non-neutral dynamics

PLOS ONE

Dear Dr. Borda-de-Agua,

Thank you for submitting your manuscript to PLOS ONE. After careful consideration, we feel that it has merit but does not fully meet PLOS ONE’s publication criteria as it currently stands. Therefore, we invite you to submit a revised version of the manuscript that addresses the points raised during the review process.

ACADEMIC EDITOR:

Thank you for submitting your work to Plos One and supporting open science. We have now received two constructive reviews of your manuscript. Both reviewers find the approach novel and interesting; both reviewers also provide critiques that must be addressed in order for the manuscript to meet Plos One publication criteria. In particular, R1 and R2 both provide useful critiques related to the assumptions of the neutral model and its application to human language distributions. The data and code for analyzing the data must also be made fully available to meet Plos One publication criteria.

We look forward to receiving your revised manuscript.

Kind regards,

Jacob Freeman

Academic Editor

PLOS ONE

Additional Editor Comments (if provided):

Reviewers' comments:

Reviewer's Responses to Questions

**Comments to the Author**

1. Is the manuscript technically sound, and do the data support the conclusions?

Reviewer #1: Partly

Reviewer #2: Partly

2. Has the statistical analysis been performed appropriately and rigorously? 

Reviewer #1: Yes

Reviewer #2: N/A

3. Have the authors made all data underlying the findings in their manuscript fully available?

Reviewer #1: Yes

Reviewer #2: No

4. Is the manuscript presented in an intelligible fashion and written in standard English?

Reviewer #1: Yes

Reviewer #2: Yes

5. Review Comments to the Author

Reviewer #1: Reviewer Comments, PONE-D-21-15698: The Relative Abundance of Languages: Neutral and Non-neutral Dynamics

Overview Comments

1. I think this is an interesting perspective on language dynamics, but I wonder if predicting the number of languages is sufficient in the absence of insights on the processes that generate more or (the current problem) fewer languages, or fewer (declining) numbers of speakers for many languages. Neutral Theory in ecology, as I understand it, relies heavily on stochastic events to create more species or remove them from a particular place. But the underlying processes are not necessarily illuminated. However, in the case of language loss, we do have a reasonable idea of the causes and it would be good to link some of these more effectively to the paper. The authors begin with the claim that many of the 7,000 languages currently on Earth may not survive this century, which I believe as well. But I think somehow connecting this study with underlying processes provides us with an idea of what we might do about this terrible loss of linguistic diversity. So, in lines 55-58 when the authors mention the similarities between the distributions of species and language abundance, I wonder if some appeal to other insights (notably on the language side) might be offered. They allude to some of these underlying processes in Lines 323-336, in the discussion of USA and Australia language distributions, but there are many of these sorts of impacts affecting Indigenous languages, a continuation of tragic processes introduce over the past few centuries..

2. Building on 1, above, the form vs. function issue is something that occurred to me when I first read the manuscript—that is, just because there are similarities between the distributions of species in ecosystems and languages in countries does not necessarily mean that the causes of distributions in the first are necessarily the same, or even similar, to the causes in the second. I do acknowledge that parameter values generate the resulting distributions, so there is a causal connection there. Ironically, I do believe that many of the processes ultimately underlying the disappearance of species are the same as those ultimately underlying the disappearance of languages—the expansion of modern economies that introduce widespread cultural and economic change as well as compromise natural habitat. But the devil is in the details, and there are so many complexities in language loss (colonization, linguistic shift where there is replacement of one language with another, migration where families comprise individuals speaking different languages that do not get passed to their children, perceived loss of importance [or value of Indigenous languages, active efforts to remove certain languages, etc.]), ultimately affecting intergenerational transmission, that it would seem quite useful to identify causes of language loss, at least as examples. The authors allude briefly to underlying causes—e.g., Colombia and the dominance of Spanish (a Colonial language)—but I think the paper would be improved if these causes were examined more extensively.

3. I suspect that half of the neutral theory approach to understanding the biological world—namely speciation—really is not happening to any appreciable extent in languages, and probably has not for several centuries.

4. It seems that the authors contend that in many countries languages are growing at similar rates, and they use Cameroon as an example (and as a place where they can modify rates to produce flattened Allen-Savage distributions) of this. As a result, the Allen-Savage distributions. But it seems to me that nearly half of the world’s ~7,000 of languages are endangered, according to Ethnologue—that is, they are not growing at the same rates as other languages (or growing at all). So, having an ability to identify areas where large numbers of languages likely emerged and the numbers of speakers (Lines 221-224 ... see discussion under Specific Comments below) would be quite interesting, but I am not sure that this method and its assumptions fit linguistic reality given that the rates of change in number of speakers of many languages quite likely vary.

Specific Comments

1. Line 61: “integer” instead of “integers”?

2. Lines 69-70: One problem is that there are many countries with a dominant language, often due to some form of colonization or overriding influence by another country. This happened less in the tropics than in more temperate areas, the latter being places where people from many European colonial powers were more comfortable (Crosby’s The Columbian Exchange discusses these broad geographic patterns). This is important because patterns of language occurrence have been affected in general by these patterns of colonial presence as well. This relates to what I mentioned in my Overview Comments 1 and 2 as a need to connect the statistical distributions with what is happening on the ground in the form of different changes and causes of those changes.

3. Line 80: Maybe clarify this assumption to state that all individuals within a given language are equal in their likelihood of transmission of a language to offspring? This varies dramatically among languages, as presented in the Ethnologue through their use of EGIDS (expanded graded intergenerational disruption scale) measures. The assumption is questionable within a language as well, but not as much as among languages.

4. Lines 84-86: Not sure how realistic this assumption is, given the vastly different geographic and sociocultural settings where various languages occur. ... so, if it is central to generating these distributions, it might be worthwhile to justify it somehow (and I am not sure that Lines 91-93 cover it, but perhaps they do).

5. Line 91: Delete “or go extinct”? I think this is clearer if arise and go extinct are counter to one another without the additional phrase.

6. Lines 115-117: As you know, many languages occur in multiple neighboring countries, the national boundaries largely being artificial (or certainly not well-grounded in sociocultural considerations). I gather that this is not the case with the biogeographic regions that the authors discuss, making me wonder if this claim of equivalence is questionable.

7. Line 122: should be “country of interest” instead of “region”?

8. Lines 122-123: I think assumption iii is quite questionable, certainly in modern times, but also associated with many centuries of colonial influence and colonization (see Specific Comment 2 above).

9. Lines 153-156 describe, briefly, many of my concerns ... certainly in the world of the 21st century, but also in many preceding centuries, where many languages extend beyond national boundaries and many areas have been exposed to a variety of impacts that can (and do) affect language richness. See comments 2, 4, 6, and 8.

10. I am not sure we completely understand the underlying causes of linguistic diversity. I think Nettle’s proposition likely holds for certain places and have made that argument myself; there are many examples that one can find in the anthropological literature. But I am not sure that it holds everywhere ... the truth is, we don’t really know. Moreover, the measure of risk in terms of length of growing season presumably assumes that the linguistic diversity you are measuring is based on agricultural economies, and in traditional times this 1) was not always the case and 2) varied broadly in terms of the amount of reliance on crop production.

11. Lines 196, 202: I’d put footnotes to define the parameters you are estimating ... I know they are defined in text, but I think it would be useful to have them here also so the reader does not have to look through previous pages to find their definitions.

12. Lines 219-220: “are readily interpreted” instead of “have readily interpretations”?

13. Lines 221-224: this is important, and I might state it in slightly different terms earlier in the paper to provide a sense of where you are heading with this entire exercise. This addresses, in part, my Overview Comment 1, where I was questioning why the authors were going through this exercise. This sort of information, on Lines 221-224, could be quite useful ... but I wonder if the questionable validity of assumptions (e.g., as noted on Lines 224-226 and in the pages that follow) really weakens the results?

14. Lines 251-255: Do Bahasa in Indonesia and Tagalog (Filipino) in the Philippines not play the role of Spanish in Colombia for your current purposes ... as both are national languages spoken by millions of people?

15. Lines 277-298: This is interesting largely because I think many linguists would contend that languages almost everywhere are growing at different rates ... or, perhaps better stated, declining (negative growth?) at different rates. My expectation is that this is happening in many places, perhaps least in Africa where one finds linguistic drift leading to main languages coming to dominate countries or portions of the continent, but where wholesale replacement is not as extreme as in much of the New World (for instance, where Spanish and English have driven out many Indigenous languages). Note that I just saw that you note the African case in Lines 318-319, and the US (with Australia ... also an instance of a colonial language replacing Indigenous languages) in Lines 323-336.

Reviewer #2: The paper provides an interesting take on the question of large-scale linguistic diversity distributions, specifically in relation to the so-call LADs (language abundance distributions). I'll jump directly to my comments and suggestions:

1) In general, the motivation for looking into NTBB for explaining linguistic distributions is very weak in the text. Some of the parallelisms are explored later in the text, but it wouldn't harm to be a bit more verbose early on.

2) Why do you write "Although the assumption of equality at individual level has been controversial in ecology (e.g., [13]), the neutrality assumption of all language speakers is less likely to generate controversy". Fertility varies greatly across human groups, and one characteristic that indexes human groups is their languages.

3) My biggest issue with your model is that, in order to meaningfully use countries as self-contained units, you need to do one of the two: I. you motivate countries as relatively isolated entities with independent dynamics or II. you show that the Allen-Savage distribution is stable. Does a mixture of A-S distributions gives rise to a A-S distribution? If not, the main argument is severely compromised.

4) The fitted Js values are most of the time really large (between 1-10k). You warn the readers about interpreting Js due to the demographic growth experienced in those countries but then - what advantage remains of the benefits of a mechanistic interpretation of A-S?

5) Code and data are not included

6. PLOS authors have the option to publish the peer review history of their article (what does this mean?). If published, this will include your full peer review and any attached files.

Reviewer #1: No

Reviewer #2: No

---

## [Author Response · Author response to Decision Letter 0]

31 Aug 2021

Reviewer #1: Reviewer Comments, PONE-D-21-15698: The Relative Abundance of Languages: Neutral and Non-neutral Dynamics

Overview Comments

1. I think this is an interesting perspective on language dynamics, but I wonder if predicting the number of languages is sufficient in the absence of insights on the processes that generate more or (the current problem) fewer languages, or fewer (declining) numbers of speakers for many languages. Neutral Theory in ecology, as I understand it, relies heavily on stochastic events to create more species or remove them from a particular place. But the underlying processes are not necessarily illuminated. However, in the case of language loss, we do have a reasonable idea of the causes and it would be good to link some of these more effectively to the paper. The authors begin with the claim that many of the 7,000 languages currently on Earth may not survive this century, which I believe as well. But I think somehow connecting this study with underlying processes provides us with an idea of what we might do about this terrible loss of linguistic diversity. So, in lines 55-58 when the authors mention the similarities between the distributions of species and language abundance, I wonder if some appeal to other insights (notably on the language side) might be offered. They allude to some of these underlying processes in Lines 323-336, in the discussion of USA and Australia language distributions, but there are many of these sorts of impacts affecting Indigenous languages, a continuation of tragic processes introduce over the past few centuries.

The general question raised in this comment, namely the suggestion to introduce potentially many possible processes that might explain the dynamics of a system, is an important issue in one’s philosophy of modeling, and therefore we think it is important to explain our general approach.

We think that attempting to model a system by including all perceived mechanisms one can conceive of is not likely to be productive, much less tractable. Instead, we prefer the approach of physicists, which is to start with simple models that are falsifiable, and to add more detail and complexity only when they become necessary to fit the data. This approach we believe is more likely to reveal the need to revise the assumptions of the model, which is exactly what happened in the present work. We believe in the value of first approximations, which is to ask, how far can one get with the simplest assumptions? 

Therefore, although we fully agree that there are potentially many processes that affect the diversity (number and relative size) of languages, we think that a first approximation that focus on a small number of variables is the most viable approach to gain insight into how to describe the system quantitatively. It is true, as the reviewer correctly points out, that neutral theory acknowledges the importance of stochastic processes and their essential importance in shaping natural patterns. However, it is incorrect to say that neutral theory does not consider processes. Neutral theory has a number of mechanistic processes, for instance, speciation and migration/dispersal. In the present case, our theory considers glossogenesis, which synthesizes a large number of (micro) processes, from social to environmental ones.

 Such macroscopic approach may not interest everyone, and we recognize that particular problems at small spatial or temporal scales may require a more detailed analysis. But there is merit also in taking a more macroscopic approach. To make our approach more concrete, an example from physics may be in order. Before the advent of statistical physics, thermodynamics had to rely on “simple” macroscopic properties of a system, such as temperature. Now we interpret temperature as a measure of the kinetic energy of the particles. This represents a deeper understanding of nature, but the lack of such understanding at the early stages of thermodynamics did not deter physicists and engineers from using the concept of temperature in theoretical and practical applications. In fact, we still do it routinely, and only resort to a more complicated description when necessary. 

2. Building on 1, above, the form vs. function issue is something that occurred to me when I first read the manuscript—that is, just because there are similarities between the distributions of species in ecosystems and languages in countries does not necessarily mean that the causes of distributions in the first are necessarily the same, or even similar, to the causes in the second. I do acknowledge that parameter values generate the resulting distributions, so there is a causal connection there. Ironically, I do believe that many of the processes ultimately underlying the disappearance of species are the same as those ultimately underlying the disappearance of languages—the expansion of modern economies that introduce widespread cultural and economic change as well as compromise natural habitat. But the devil is in the details, and there are so many complexities in language loss (colonization, linguistic shift where there is replacement of one language with another, migration where families comprise individuals speaking different languages that do not get passed to their children, perceived loss of importance [or value of Indigenous languages, active efforts to remove certain languages, etc.]), ultimately affecting intergenerational transmission, that it would seem quite useful to identify causes of language loss, at least as examples. The authors allude briefly to underlying causes—e.g., Colombia and the dominance of Spanish (a Colonial language)—but I think the paper would be improved if these causes were examined more extensively.

We believe we have partially answered this comment before. As we previously acknowledged, a “macroscopic” approach may not be enough to understand or solve a given problem, and in this sense we fully agree that the “the devil is in the details”. However, our understanding of this field (the dynamics/demography of languages) is that it is still in its infancy. If we still have a poor understanding of the system, starting by using a very complicated model does not seem to us to be the best approach. 

The two main parameters of our model are size of the population at the origination and “speciation” rate, are determined by several processes; for instance, a high “speciation” rate may be related to the local topography. As we mentioned before, these two variables “aggregate” several mechanisms. Starting by realizing their different values among different regions can be a starting point (and a guidance) to develop a deeper look at the processes causing those differences. In other words, obtaining the quantitative values for these parameters can be seen as an indispensable first step that will guide the search for the underlying processes. In this sense, we do not agree with the referee when he/she states “I think the paper would be improved if these causes were examined more extensively”, because our main purpose was to start by introducing a relatively simple model (since it has only a small number of assumptions). As the reviewer pointed out in some other comments, we did refer to some possible microscopic causes underlying the languages dynamics, and we did this precisely because our model does not always work, which, in fact, we see as a success. However, to discuss other processes is, we think, outside the scope of this paper.

Obviously, when applying a model, care is required. Although we dealt with “macroscopic” variables, we recognize that these are emergent properties of the underlying processes, and these may differ considerably among different societies and geographical regions. This is the reason why we use countries as the unit of study, and remove those that were not sufficiently homogeneous as measured by the habitat risk variable. This is not ideal, but it was the best we could do with the present data and information.

3. I suspect that half of the neutral theory approach to understanding the biological world—namely speciation—really is not happening to any appreciable extent in languages, and probably has not for several centuries.

We agree, this may well be the case (but obviously it had to happen at same point).

4. It seems that the authors contend that in many countries languages are growing at similar rates, and they use Cameroon as an example (and as a place where they can modify rates to produce flattened Allen-Savage distributions) of this. As a result, the Allen-Savage distributions. But it seems to me that nearly half of the world’s ~7,000 of languages are endangered, according to Ethnologue—that is, they are not growing at the same rates as other languages (or growing at all). So, having an ability to identify areas where large numbers of languages likely emerged and the numbers of speakers (Lines 221-224 ... see discussion under Specific Comments below) would be quite interesting, but I am not sure that this method and its assumptions fit linguistic reality given that the rates of change in number of speakers of many languages quite likely vary.

This is a fair comment, although we would slightly rephrase it. We would rather say that the good fit provided by the Allen-Savage distribution is compatible with populations (identified by their languages) growing at similar rates. Stated in this way, this is a prediction of the model that could in principle be tested with real data. Unfortunately, we do not know of such detailed data. In the Supplementary material (S4), we addressed some questions that can be dealt with our model, but we did not include them in main text because the data are of very poor quality. Still, we think that our model and future versions of it, together with other methods, such as ones borrowed from population genetics, may in the future shed some light on the issues of language origination and dynamics.

Specific Comments

1. Line 61: “integer” instead of “integers”?

Thank you for pointing out this mistake. We have corrected it.

2. Lines 69-70: One problem is that there are many countries with a dominant language, often due to some form of colonization or overriding influence by another country. This happened less in the tropics than in more temperate areas, the latter being places where people from many European colonial powers were more comfortable (Crosby’s The Columbian Exchange discusses these broad geographic patterns). This is important because patterns of language occurrence have been affected in general by these patterns of colonial presence as well. This relates to what I mentioned in my Overview Comments 1 and 2 as a need to connect the statistical distributions with what is happening on the ground in the form of different changes and causes of those changes.

The reviewer is obviously correct about the importance of colonization or the influence of other countries. We recognize this when we mentioned Colombia, and the different fit obtained with and without the inclusion of Spanish (Fig. 1 caption). Clearly, when there is an overwhelming influence of a language our model does not provide a good description of the system, as revealed by the poor fitting given to the distribution of Colombia. Again, the poor fit in some cases and the good fit in other is an important lesson that we can take from the model.

We would like to reiterate some of the points made before. We believe that when we start quantitatively modelling a (complex) system, it is advisable to start with a simple system, see when it works and, crucially, identify when it does not work, and then add detail as required. Our model works well for some language systems, but not in others, which points future researchers to which language systems need more detailed causal explanations. Theory is always a work in progress. What we have done here is present a model to the scientific community that explains some of the dynamics of language diversity and also illuminates cases where the simplest model does not work. We hope this work will serve as a useful guide to the scientific community about which avenues are likely to be most productive in advancing our understanding of the dynamics of language diversity. 

It would be rather naïve to assume that there were no situations in the past where there was a dominant language. For instance, the expansion of Latin and the replacement of Celtic and other languages, comes to mind. So, we envisage our model to work better when there isn’t a dominant group that, forcibly or not, imposes a language.

3. Line 80: Maybe clarify this assumption to state that all individuals within a given language are equal in their likelihood of transmission of a language to offspring? This varies dramatically among languages, as presented in the Ethnologue through their use of EGIDS (expanded graded intergenerational disruption scale) measures. The assumption is questionable within a language as well, but not as much as among languages.

This comment is similar to comment 2 of reviewer 2. The assumption of all individuals being equal is obviously wrong. But the questions are: is it a good first approximation? How much can we learn by assuming it? We argue that under similar conditions (economic, environmental, etc.) it is a reasonable first approximation to assume that individuals are equal, independently of their languages. However, two points are worth emphasizing. First, assuming similar environmental conditions is also important, and that is the reason why we apply our model at the country level. Countries are not perfectly uniform environmentally or economically, but they are more so than had we adopted multi-national units. Among countries, only a few were relatively homogeneous based on the “ecological risk” hypothesis (which again is far from perfect, see below answer to comment 10). Second, these idealized similar conditions may have been closer to those of pre-industrial societies, and not such a good approximation for the present societies, with some exceptions. We agree that our model may be better at reproducing conditions in the past, and thus of value for those studying pre-industrial (or even pre-historic societies), than at describing the conditions prevalent in modern societies. 

In order to clarify this issue, we added the following sentences (the new text is in italic):

“Although the assumption of equality at individual level has been controversial in ecology (e.g., [13]), the neutrality assumption of all language speakers is less likely to generate controversy, under the assumption that all individuals are broadly subject to the same social and environmental conditions, in line with the applications of NTBB in ecology where it is assumed that all individuals belong to the same trophic level and the same biogeographic region (e.g., the Amazon basin).” 

“We used data on languages and number of speakers per country from the Ethnologue website [4]. The reason for choosing country as the unit of analysis (and not, say, continental or global scales) is to ensure that individuals are subject to similar conditions. However, this leaves the question of the environmental heterogeneity within a country, and how this heterogeneity affects the in-country linguistic diversity, unanswered.”

4. Lines 84-86: Not sure how realistic this assumption is, given the vastly different geographic and sociocultural settings where various languages occur. ... so, if it is central to generating these distributions, it might be worthwhile to justify it somehow (and I am not sure that Lines 91-93 cover it, but perhaps they do).

In line with our previous answers, the assumption that all languages start with the same number of individuals within a given region (country for the purposes of the paper) is a first approximation. We emphasize that the “same number of individuals” depends on the country/region considered). Previous studies (e.g., Binford et al. 1968; Birdsell et al. 1973), on minimum size of a human group (Binford et al. 1968; Birdsell et al. 1973) have arrived at numbers that are not terribly different, within the same order of magnitude.

In order to clarify this issue, we added the following sentences (the new text is in italic):

“There are several versions of the NTBB depending on the speciation (or glossogenesis) mode adopted [10-11]; and see Supporting Information S3. For present purposes, we assume that each incipient language (within a homogeneous region) starts with the same fixed number of individuals. Clearly, the assumption of the same starting number of individuals is a first approximation, but in broad agreement with previous estimates of initial population sizes [12].” 

Binford, L., Birdsell, J. B., Damas, D., Freeman, L., Hiatt, L., Sahlins, M., & Washburn, S. (1968). The magic numbers" 25" and" 500": Determinants of group size in modern and Pleistocene hunters. Man the hunter, 245-248.

Birdsell, J. B., Bennett, J. W., Bicchieri, M. G., Claessen, H. J. M., Gropper, R. C., Hart, C. W. M., ... & Thompson, L. (1973). A basic demographic unit [and comments and reply]. Current Anthropology, 14(4), 337-356.

5. Line 91: Delete “or go extinct”? I think this is clearer if arise and go extinct are counter to one another without the additional phrase.

Thank you for pointing this to us. Obviously “or go extinct” should not be part of the sentence.

6. Lines 115-117: As you know, many languages occur in multiple neighboring countries, the national boundaries largely being artificial (or certainly not well-grounded in sociocultural considerations). I gather that this is not the case with the biogeographic regions that the authors discuss, making me wonder if this claim of equivalence is questionable.

This is an excellent point with which we debated considerably. For those languages occurring in multiple neighbouring countries it could make more sense to consider them all together as part of a given region. On the other hand, the problem is that groups in different countries speaking the same languages can be under very different conditions. Therefore, we concluded that it is better to assume the country as the unit of study. However, we do acknowledge that this is a difficult issue, to which we do not have a ready solution. We added the following text, recognizing the importance of this issue:

“Clearly, countries are not closed systems; migrations occur and often the same language is spoken in different countries. The advantage of using a country as the basic spatial unit is that we consider, as a first approximation, that its linguistic populations are subjected to the similar environmental and social conditions. On the other hand, we assume that if migrations occur these do not have a strong impact in the overall language abundance distribution; if this is not the case, such is the case in periods of social upheaval, the model we now describe will not apply.”

7. Line 122: should be “country of interest” instead of “region”?

Yes, it should be “country of interest”. We corrected it. Thank you.

8. Lines 122-123: I think assumption iii is quite questionable, certainly in modern times, but also associated with many centuries of colonial influence and colonization (see Specific Comment 2 above).

We agree that assumption iii (the total origination rate of languages equals the extinction rate) may not apply to modern times, after colonization or periods of colonial influence. This assumption is used to imply an equilibrium during which the number of languages remains the same. We do not believe that a “perfect” equilibrium ever existed but that it may be a good approximation during some historical periods, and that it may have “shaped” the language abundance distributions which have been changing in more recent times (including in some countries after colonization).

We added the following text:

“For instance, assumption (iii) implies that the number of languages is constant over time, and we know that at present a large number of languages is becoming extinct. Equally, assumption (v) does not hold at the present given the fast growth of most human populations. We will discuss their broad implications in due time.”

9. Lines 153-156 describe, briefly, many of my concerns ... certainly in the world of the 21st century, but also in many preceding centuries, where many languages extend beyond national boundaries and many areas have been exposed to a variety of impacts that can (and do) affect language richness. See comments 2, 4, 6, and 8.

We think we addressed this issued in our previous answers.

10. I am not sure we completely understand the underlying causes of linguistic diversity. I think Nettle’s proposition likely holds for certain places and have made that argument myself; there are many examples that one can find in the anthropological literature. But I am not sure that it holds everywhere ... the truth is, we don’t really know. Moreover, the measure of risk in terms of length of growing season presumably assumes that the linguistic diversity you are measuring is based on agricultural economies, and in traditional times this 1) was not always the case and 2) varied broadly in terms of the amount of reliance on crop production.

We completely agree that we do not “completely understand the underlying causes of linguistic diversity” and we hope that we did not leave such impression in the text. It is also unlikely that the hypothesis of ecological risk holds everywhere. And we also think that it is especially no true at the present time in the majority of the countries, where the variability of food production could be decoupled from major upheavals in the society. However, as the reviewer mentioned above “namely speciation—really is not happening to any appreciable extent in languages, and probably has not for several centuries” most languages (thus their diversity) originated in periods where most societies were based on agriculture, therefore, we argue that using ecological risk, or another index that is related to agriculture, or more broadly, environmental conditions, is a reasonable assumption. 

11. Lines 196, 202: I’d put footnotes to define the parameters you are estimating ... I know they are defined in text, but I think it would be useful to have them here also so the reader does not have to look through previous pages to find their definitions.

Thank you for this recommendation. We added the definition of the parameters to the tables’ captions.

12. Lines 219-220: “are readily interpreted” instead of “have readily interpretations”?

Indeed, “are readily interpreted” is much better.

13. Lines 221-224: this is important, and I might state it in slightly different terms earlier in the paper to provide a sense of where you are heading with this entire exercise. This addresses, in part, my Overview Comment 1, where I was questioning why the authors were going through this exercise. This sort of information, on Lines 221-224, could be quite useful ... but I wonder if the questionable validity of assumptions (e.g., as noted on Lines 224-226 and in the pages that follow) really weakens the results?

We added to the introduction the following sentence following this recommendation (new text in italics). 

“The two parameters of the Allen-Savage distribution are JS, the incipient size of the population, and �, the per capita glossogenesis (speciation) rate, both capturing important characteristics of language dynamics. In fact, estimating these parameters can help identify regions in the globe with different rates of species origination and the typical size of a human group. This observation is important because the bell-shape of the LADs…”

14. Lines 251-255: Do Bahasa in Indonesia and Tagalog (Filipino) in the Philippines not play the role of Spanish in Colombia for your current purposes ... as both are national languages spoken by millions of people?

Indeed, Bahasa in Indonesia and Tagalog (Filipino) in the Philippines play a role similar to that of Spanish in Colombia. Notice, however, that in our sentence 

“[in] Indonesia and Philippines (Fig. 1i-k) there is not a language with a much larger (isolated) number of speakers”

 we refer to the fact that one language is much more “abundant”. Inspecting the distributions of language abundance in Indonesia and the Philippines, one sees that the largest language occupies a bin that is adjacent to other occupied bins, whereas in the case of Colombia there is a “gap” between the bin occupied by Spanish and the other occupied bins, i.e., Spanish is much larger in abundance than the other languages in Columbia. We chose Colombia in addition to Indonesia and the Philippines to illustrate a case where there is such a huge different between the most spoken language and the others.

15. Lines 277-298: This is interesting largely because I think many linguists would contend that languages almost everywhere are growing at different rates ... or, perhaps better stated, declining (negative growth?) at different rates. My expectation is that this is happening in many places, perhaps least in Africa where one finds linguistic drift leading to main languages coming to dominate countries or portions of the continent, but where wholesale replacement is not as extreme as in much of the New World (for instance, where Spanish and English have driven out many Indigenous languages). Note that I just saw that you note the African case in Lines 318-319, and the US (with Australia ... also an instance of a colonial language replacing Indigenous languages) in Lines 323-336.

We agree with the reviewer observations. In fact, what the reviewer points out in here are some of the main conclusions of our work, see lines 384:404. However, in practice, we do not believe that there are any situations where all the languages have the same (positive or negative) growth rate. We just contend that in some cases languages may have similar growth rates.

Reviewer #2: The paper provides an interesting take on the question of large-scale linguistic diversity distributions, specifically in relation to the so-call LADs (language abundance distributions). I'll jump directly to my comments and suggestions:

Some of the comments by reviewer 2 are similar to those by reviewer 1. Because we started by answering reviewer 1, in addition to our replies below we refer to some of our previous ones.

1) In general, the motivation for looking into NTBB for explaining linguistic distributions is very weak in the text. Some of the parallelisms are explored later in the text, but it wouldn't harm to be a bit more verbose early on.

Thank you for this observation. In order to elaborate on the motivation of using NTBB we added the following sentences to the introduction (new text is italics):

“Furthermore, subjects in ecological and linguistic communities are subjected to similar processes, such as, birth, death, and probability of speciating (ecology) or giving rise to a new language (linguistics). Therefore, the similarities between the patterns predicted by the NTBB and language abundance distributions, and the similarity of the underlying processes, make the NTBB a natural candidate to test hypotheses about when language dynamics might be governed by neutral or non-neutral processes.”

2) Why do you write "Although the assumption of equality at individual level has been controversial in ecology (e.g., [13]), the neutrality assumption of all language speakers is less likely to generate controversy". Fertility varies greatly across human groups, and one characteristic that indexes human groups is their languages.

This comment is similar to “specific comment” 3 of reviewer 1. The neutrality assumption is being made at the individual level, while the argument raised here is at the “human groups” level, or probably one could say at the population level. What is implicit in our application of neutral theory to this particular case is that individuals under the (approximately) same conditions are identical in their per capita rates of “reproducing (transmitting the language to offspring), dying and giving rise to a new language”. The issue raised here, is related to the next one. We chose countries of unities of study basically because that was the smallest unit for each we had information, but because we recognised that we may have tremendous variation within a country, we removed several ones from our analyses, as explained in lines 202:220. Of course, human populations even when not distinguished by their languages, may have different fertility rates (for example, different religious groups). In order to clarify this issue, we added the following sentences (the new text is in italic):

“Although the assumption of equality at individual level has been controversial in ecology (e.g., [14]), the neutrality assumption of all language speakers is less likely to generate controversy, under the assumption that all individuals are broadly subject to the same social and environmental conditions. This is in line with the applications of NTBB in ecology where it is assumed that all individuals belong to the same trophic level and the same biogeographic region (e.g., the Amazon basin).” 

“Here we treat each country as a self-contained unit where the processes of death, birth and glossogenesis occur; thus, a country is the equivalent of a biogeographic region in the original NTBB. The purpose of choosing country as the unit of analysis (and not as, say, continental or global scale) is to ensure that individuals are subject to similar conditions.”

“We used data on languages and number of speakers per country from the Ethnologue website [4]. The reason for choosing country as the unit of analysis (and not, say, continental or global scales) is to ensure that individuals are subject to similar conditions. However, this leaves the question of the environmental heterogeneity within a country, and how this heterogeneity affects the in-country linguistic diversity, unanswered.”

3) My biggest issue with your model is that, in order to meaningfully use countries as self-contained units, you need to do one of the two: I. you motivate countries as relatively isolated entities with independent dynamics or II. you show that the Allen-Savage distribution is stable. Does a mixture of A-S distributions gives rise to a A-S distribution? If not, the main argument is severely compromised.

We cannot prove that a mixture of A-S distributions gives rise to a A-S distribution, but we believe this is unlikely. On the other hand, we think countries can be seen (as a first approximation) as isolated with independent dynamics during some periods of time. Obviously, this is not always the case. Our argument is that realistically the language dynamics only follows the assumptions of our model during some periods of time. Major upheavals may considerable change the language community and, evidently, during those periods the assumption of our model do not apply (which is probably what is happening at the present in most countries). It is likely that small migrations do occur all the time but when we consider distributions with a large number of languages, such small migrations are unlikely to cause major changes in the distributions (see reply to “specific comment# 6 by reviewer 1).

4) The fitted Js values are most of the time really large (between 1-10k). You warn the readers about interpreting Js due to the demographic growth experienced in those countries but then - what advantage remains of the benefits of a mechanistic interpretation of A-S?

The benefits remain because if one has enough data to reconstruct the linguistics groups growth one could in principle obtain the original size of Js at the time of origin. In fact, we briefly explored this in the appendix S.4. As we mention in the reply to the “overview comment 4” by reviewer 1, we did not include these analyses in the main text because we are not confident about the quality of the data. As we say in appendix S.4 this was done “mainly to show the potential of applying our approach to obtain a better understanding of the historical dynamics of language origination.” 

5) Code and data are not included

We have now included the data, the codes and an explanation of how to use them. However, notice that we have some issues concerning the use of the data, that we would like to clarify with PLOS ONE. Namely, we downloaded the data from the Ethnologue when it was freely available, but since that is no longer the case, we are not sure on how to deal with the issue of providing the data we used. Since permission to freely use the data is no longer the current policy of Ethnologue, we defer questions to the providers of these data about how these data can be accessed.

---

## [Decision Letter · Decision Letter 1]

14 Oct 2021

The relative abundance of languages: neutral and non-neutral dynamics

PONE-D-21-15698R1

Dear Dr. Borda-de-Agua,

We’re pleased to inform you that your manuscript has been judged scientifically suitable for publication and will be formally accepted for publication once it meets all outstanding technical requirements.

Kind regards,

Jacob Freeman

Academic Editor

PLOS ONE

Additional Editor Comments (optional):

Reviewers' comments:

Reviewer's Responses to Questions

**Comments to the Author**

1. If the authors have adequately addressed your comments raised in a previous round of review and you feel that this manuscript is now acceptable for publication, you may indicate that here to bypass the “Comments to the Author” section, enter your conflict of interest statement in the “Confidential to Editor” section, and submit your "Accept" recommendation.

Reviewer #1: (No Response)

2. Is the manuscript technically sound, and do the data support the conclusions?

Reviewer #1: Partly

3. Has the statistical analysis been performed appropriately and rigorously? 

Reviewer #1: Yes

4. Have the authors made all data underlying the findings in their manuscript fully available?

Reviewer #1: Yes

5. Is the manuscript presented in an intelligible fashion and written in standard English?

Reviewer #1: Yes

6. Review Comments to the Author

Reviewer #1: Reviewer Comments, PONE-D-21-15698: The Relative Abundance of Languages: Neutral and Non-neutral Dynamics

This paper continues, in its revised version, to encounter some of the same problems that I pointed to in my initial review. I read the response to comments of both reviewers twice, along with the revised manuscript. There seem to be some similarities in reviewer comments, and it may have been more prudent to try to address those comments by revising the manuscript rather than providing responses to reviewers (and, presumably, the editors) arguing the utility of simple models and first approximations.

I think this version of the paper reads better than the first version, but I continue to have some problems with the overall product. The two parameters of the Allen-Savage distribution are incipient size and rate of glossogenesis/speciation—so how many you start off with and the rate at which new versions appear having an effect on the number of languages and number of speakers of each. It is tough to argue against that. I would agree that this is a simple model and, and I suppose in this simplicity it would apply to species as well as languages (as well as perhaps other things ... again, a benefit of its simplicity). Adding several assumptions that help make the modeling process more feasible but, as the authors acknowledge, are unrealistic or untrue, in my mind compromises any noteworthy understanding that the model might contribute. And yes ... I understand that this can be an important part of the modeling process.

In pointing out the utility in considering complicating factors in my initial review, I was not suggesting that the authors account for every possible condition affecting every language included in the study, which clearly would be impossible. But there are major processes that have occurred that have affected languages—the numbers of languages that occur per country and the numbers of speakers of each, as reported in Ethnologue—that should be addressed to help explain the languages and speakers that exist. One such major process is the colonization that brought Spanish to Colombia (an example in the paper) and many other parts of Latin America. As patterns of colonization occurred differently in different parts of the world, such that large sections of the tropics were of less interest to European colonial powers than more familiar temperate areas (the Crosby reference in the initial review), one might ask how those played out in terms of the Allen-Savage results or deviations from it. Have levels of industrialization, amounts of natural habitat destruction, differences in human well-being (economic, health, etc.), and so on played a role in observed fits or lack thereof for the model. Again, to be clear, I am not (and was not) advocating that the authors get buried in the minutia, but rather step back and consider some of the broad processes or measures of differences that affect the numbers of languages and the numbers of speakers. Otherwise, I think you arrive at a larger incipient size and a faster glossogenesis/speciation rate generate a large number of languages and a large number of speakers, which is hard to argue against.

I appreciate that there has been a large amount of thought and work has gone into this analysis and the resulting manuscript. And I agree that simple models are a good way to start out. Certainly such models facilitate monitoring the effect of parameters on results and tracking how invalidating assumptions can modify output. I guess in my way of thinking, I would begin with the model results under ideal conditions (which the authors do), but then spend more time trying to understand causes of deviation from those results—causes that, I suspect, may well play out geographically, and which would provide a more powerful statement about why we observe the numbers of languages and the numbers of speakers reported in Ethnologue.

7. PLOS authors have the option to publish the peer review history of their article (what does this mean?). If published, this will include your full peer review and any attached files.

Reviewer #1: No

---

## [Editor Report · Acceptance letter]

2 Dec 2021

PONE-D-21-15698R1 

The relative abundance of languages: neutral and non-neutral dynamics 

Dear Dr. Borda-de-Agua:

I'm pleased to inform you that your manuscript has been deemed suitable for publication in PLOS ONE. Congratulations! Your manuscript is now with our production department. 

Kind regards, 

on behalf of

Dr. Jacob Freeman 

Academic Editor

PLOS ONE